# French Translation and Validation of the Ontological Addiction Scale (OAS)

**DOI:** 10.3390/ijerph22040511

**Published:** 2025-03-27

**Authors:** Déborah Ducasse, Martin Leurent, Marie-Christine Picot, Safa Aouinti, Véronique Brand-Arpon, Philippe Courtet, Paul Barrows, Edo Shonin, Supakyada Sapthiang, Emilie Olié, William Van Gordon

**Affiliations:** 1Institute of Functional Genomics, National Center for Scientific Research, National Institute of Health and Medical Research (INSERM), University of Montpellier, 34094 Montpellier, France; d-ducasse@chu-montpellier.fr (D.D.); philippe.courtet@umontpellier.fr (P.C.); e-olie@chu-montpellier.fr (E.O.); 2Department of Emergency Psychiatry and Post Acute Care, Centre Hospitalier Regional Universitaire, 34090 Montpellier, France; 3Therapy Center for Mood and Emotional Disorders, Department of Adult Psychiatry, La Colombière, Centre Hospitalier Universitaire, 34090 Montpellier, France; martinleurent606@gmail.com (M.L.); vbrand.arpon@gmail.com (V.B.-A.); 4Clinical Research and Epidemiology Unit (Public Health Department), Centre Hospitalier Universitaire de Montpellier, University of Montpellier, 34295 Montpellier Cedex 5, France; mc-picot@chu-montpellier.fr (M.-C.P.); s-aouinti@chu-montpellier.fr (S.A.); 5Centre de Recherche en Épidémiologie et Santé des Populations (CESP), U1018 INSERM, Université Paris-Saclay, 94807 Villejuif, France; 6School of Psychology, University of Derby, Kedleston Road, Derby DE22 1GB, UK; w.vangordon@derby.ac.uk; 7Awake to Wisdom Centre for Meditation and Mindfulness Research, 4 Derby Grove, Nottingham NG7 1ND, UK; e.shonin@awaketowisdom.co.uk (E.S.); supakyadafilm@gmail.com (S.S.)

**Keywords:** ontological addiction, ontological addiction scale, scale validation, Buddhist-derived practices

## Abstract

According to ontological addiction theory, the root cause of mental suffering is a dysfunctional conceptualisation of the self. Typically, an individual with such a flawed self-concept deems themselves to be intrinsically separate from their surroundings, with their beliefs, choices and behaviours structured in order to reinforce their sense of an independently existing self. The Ontological Addiction Scale (OAS) was developed to measure ontological addiction and demonstrates good psychometric properties for the original version validated in English. The present study aimed to validate a French language version of the OAS. The 24-item OAS was administered to 492 French adults with emotional and/or mood disorders. The French OAS demonstrated good internal consistency (Cronbach’s alpha: 0.89) and strong test–retest reliability. We suggest a single-factor structure, aligning with the original English version. The 12-item OAS also showed good internal consistency (Cronbach’s alpha: 0.81). Construct validity was confirmed by medium to large correlations with self-esteem, shame, perfectionism and mindfulness. These results support the use of the French OAS in research and clinical practice, offering a robust measure for assessing ontological addiction as well as a dimensional assessment of psychiatric symptoms.

## 1. Introduction

“The time to include dimensional diagnosis is now…”: this is the way the DSM Taskforce concluded a meeting in 2007 [1]. Following this, Hyman [2] underlined how disorders within the DSM are often treated as if they were real entities existing independently of any particular rater, and that comorbidity is very common among DSM diagnoses. More recently, Krueger et al. [3] highlighted the extent to which the categories of official nosologies are out-of-sync with data on the dimensional nature of psychopathology, and that contemporary empirical approaches are moving toward the dimensional conceptualisation of psychopathological constructs. Furthermore, Zandersen and Parnas [4] asserted that since DSM-III, psychiatric diagnoses have become reified and considered as natural kinds, reinforcing the need to reinstate theoretical and empirical psychopathology at the core of scientific psychiatry.

Moving beyond a reductionist view of existing diagnostic categories [2], ontological addiction theory (OAT) is a metaphysical model of human functioning and mental illness [5]. This comprehensive cross-disciplinary model asserts that human beings are prone to forming implausible beliefs concerning the manner in which they exist, and that these beliefs can become addictive leading to functional impairments and mental illness [5]. The root of ontological addiction (OA) stems from a deep-rooted belief in an inherently existing “Self” or “I”: a solid entity, independent, permanent, separate from the rest of the world, existing of itself (independent of “My” mind and other phenomena) [5]. Such a belief is asserted to foster self-grasping along with a profound feeling of separation from others, dissatisfaction, insecurity, and a sense of lacking personal value [6].

According to OAT, experiences arising from this form of self-identification are ultimately grounded in a flawed view of reality [7], with the constant need to satisfy, secure, or give value to this incorrect way of relating to oneself. OAT asserts that a correct self-conceptualisation is the main factor promoting a mind that is peaceful, calm and happy, which in turns give rise to functionally adaptive behaviours. Conversely, an incorrect conceptualisation of the self serves as the main cause of making the mind chaotic, agitated and unhappy, which in turn elicits dysfunctional behaviours. Furthermore, an incorrect self-conceptualisation eventually results in a form of ontological addiction, or addiction to an incorrect conceptualisation of the self [5].

Some third-wave cognitive behavioural therapies specifically integrate Buddhist-derived practices that aim to dislodge these incorrect self-concepts, in favour of more realistic ones that promote resilient, compassionate, and creative psychological processes. Furthermore, a variety of recent therapies incorporate many closely related Buddhist practices such as mindfulness [8,9], compassion [10], and nonattachment [11,12]. This latter practice has been the source of much study in relation to addressing the ego-attachment that is the focus of OA, with a substantial growth of interest in exploring nonattachment-to-self as a measurable construct [13] and utilising this as a tool in positive personal growth and transformation [14]. Another approach to undermining the flawed beliefs underlying OA is through ‘non-dual’ practices [15] such as emptiness (Sanskrit: śūnyatā) meditation, whose purpose is to cultivate deeper states of consciousness that promote understanding of how we exist [16,17].

The Ontological Addiction Scale (OAS), which has been validated in English [18] and Turkish [19] and shows good psychometric properties [18,19], aims to quantify the severity of an individual’s dysfunctional ego-centredness and allow improvement of this fundamental psychological dimension to be tracked over the course of therapeutic intervention. The OAS was developed based on Griffiths’ [20] components model of addiction, which identifies six fundamental elements common to all addiction types: salience, mood modification, tolerance, withdrawal, conflict, and relapse. The OAS is also grounded in the Buddhist philosophical standpoint of the benefits of avoiding extremes and remaining non-attached to self and worldly happenings, as delineated in Nagarjuna’s exposition of the eight mundane concerns [21]:Feeling pleased or delighted due to having money and/or material possessions,Feeling disappointed, upset or angry due to losing possessions or not acquiring them,Feeling pleased when praised or approved of by others,Feeling upset or dejected when criticised or subjected to disapproval,Feeling pleased due to having a good reputation,Feeling dejected or upset due to having a bad reputation,Feeling delighted when experiencing sense pleasures,Feeling dejected and upset by unpleasant sensory experiences.

These eight mundane concerns embody the positive and negative aspects of four underlying components (Table 1; material wealth or possessions; sensations; reputation amongst kin and social circles; and wider reputation), with two polarities: gain and loss [18]. These facets are not separate empirical components but thematic guidelines.

In the initial scale development study, 24 specific items corresponding to each of the six addiction components were generated (see Table 2). To refine and enhance the scale’s robustness, seven additional items were included for potential replacement or to increase breadth. These items were evaluated and adjusted based on their performance in initial tests, leading to the final version featuring 24 items. Furthermore, the scale was condensed into a 12-item short form version (OAS-12), selecting items that most accurately captured the essence of ontological addiction. For both the 12- and 24-item version, respondents rate each item on a frequency scale ranging from “Never” (0) to “Always” (4). The total score, representing the severity of ontological addiction, is calculated by summing the item scores.

Although the original version of the OAT was validated in English, it was constructed through a collaborative international effort, ensuring its relevance and applicability across both English- and French-speaking populations. This involved co-creating items that accurately captured the six addiction components while considering the four high-level elements of the eight mundane concerns. The best simultaneous English and French formulations were identified to effectively reflect the targeted dimensions. This cross-cultural adaptation process, with its simultaneous bilingual development, was crucial for ensuring the scale’s linguistic accuracy and cultural resonance, to maximise its suitability for use in culturally diverse research and clinical contexts.

The primary objective of the present study was to examine the psychometric properties of the OAS specifically in a French context, using individuals experiencing emotional or mood-related psychiatric disorders. More specifically, we aimed to determine whether the French language version of OAS effectively captures the nuances of ontological addiction as it pertains to incorrect self-conceptualisation and its associated psychological attributes.

## 2. Study and Methodology

### 2.1. Study Design

This study followed a cross-sectional design, with data collected at a single time point. Ethical approval for the study was provided by the Montpellier Academic Hospital Research Ethics Committee, France. All participants provided informed consent.

### 2.2. Participants

All participants were patients at the Therapy Centre for Mood and Emotional Disorders and Department of Emergency Psychiatry and Post Acute Care, Academic Hospital of Montpellier, France. This study was advertised to all French-speaking patients within the above setting who had a current diagnosis of emotional and/or mood disorders, as determined by clinician assessment. The sample size (*n* = 492) was, therefore, determined by the number of patients who accepted the invitation to participate, with findings from the Factor Analysis being used to confirm that the sample size was adequate. Eligibility criteria were French speakers aged between 18 and 80 years old, diagnosed with mood disorder (bipolar or unipolar) and/or emotional disorder (subjective complaint for difficulties to regulate emotions).

### 2.3. Procedure

As previously noted, the 24-item version of the OAS was concurrently co-developed in English and French to ensure a linguistically and conceptually equivalent measure for both languages. In the present study, the French version of the OAS (see Appendix C) was administered to participants, who also completed several other validated measures to assess convergent and divergent validity (hypothesis testing). This step was taken following the Consensus-based Standards for the Selection of Health Measurement Instruments (COSMIN) guidelines to ensure methodological rigor and quality in psychometric evaluation [22]. A total of 492 participants completed the OAS and the other measures at two separate time points, with a 15-day interval between administrations in order to evaluate the reliability of the OAS, particularly its test–retest reliability. A 15-day test–retest interval was chosen to allow sufficient time for potential variability in state-like constructs, aiming to robustly assess the stability of the OAS over time.

Given that ontological (OA) addiction refers to the erroneous mental construction of how we believe we exist, we anticipated a negative correlation of OA with self-esteem, which was measured using the Rosenberg Self-Esteem Scale (RSES; [23]). The RSES is a 10-item scale that assesses self-esteem (e.g., “On the whole, I am satisfied with myself” and “At times, I think I am no good at all”). Participants rate items on a four-point Likert scale (“strongly agree”, “agree”, “disagree” and “strongly disagree”). Scores range from 0 to 30, with higher scores indicating higher levels of self-esteem. The psychometric properties of the RSES were validated in a French population with a Cronbach’s alpha of 0.70 [24].

We also anticipated a positive correlation of OA with the shame-proneness subscale of the Test of Self-Conscious Affect-3 Scale (TOSCA-3; [25]), because shame is a core emotion of the perceived inadequacy of an erroneous self-concept. This sub-scale comprises 16 of the 69 items of the TOSCA-3, in the form of 16 relational scenarios, accompanied by suggested reactions to these situations. Respondents are asked to evaluate each proposed reaction based on a five-point Likert scale (varying from “Not at all in agreement with my reaction” to “Completely in line with my reaction”). Shame subscale scores range from 0 to 64, with a higher score indicating higher levels of shame-proneness. The psychometric properties of the TOSCA-3 shame subscale (TOSCA-3-SS) were validated in a French population with a Cronbach’s alpha of 0.72 [26].

Furthermore, because ontological addiction implies fixation on rigid ideas about how things “must” be and the attempt to control external reality, we anticipated a positive correlation between OA and the Self-Oriented Perfectionism subscale of the Multidimensional Perfectionism Scale (SOP-MPS; [27]). The SOP-MPS is a 15-item scale that assesses self-oriented perfectionism on a seven-point Likert scale (varying from “strongly disagree” to “strongly agree”). The SOP-MPS scores range from 0 to 90, with higher scores indicating higher levels of perfection standards toward oneself, involving strict self-criticism and a tendency to constantly strive for perfection. In the original validation study for the SOP-MPS, the Cronbach alpha was 0.86. The psychometric properties of the MPS have subsequently been validated in a French population [28].

We also expected OA to be negatively correlated with acceptance of events and experiences in the present moment, as measured by the Mindful Attention Awareness Scale (MAAS; [29]). The MAAS is a 15-item scale that assesses attention and awareness in daily life on a six-point Likert scale (varying from “almost always” to “almost never”). The MAAS is composed of 15 items formulated in an indirect way (e.g., “I rush through activities without being really attentive to them”) that address cognitive, emotional, physical, interpersonal, and general trait mindfulness domains. The MAAS scores range from 0 to 50 with higher scores reflecting greater present moment awareness states. The psychometric properties of the MAAS have been validated in a French population with a Cronbach’s alpha of 0.84 [29].

Finally, we expected OA to have a negative correlation with gratitude as measured using the Gratitude Scale Questionnaire—Six Item Form (GQ-6; [30]). The GQ-6 is a six-item scale that measures the tendency to appraise, recognise and respond to life events through being grateful, based on a seven-point Likert scale (varying from “strongly disagree” to “strongly agree”). The GQ-6 scores range from 0 to 36 with higher scores reflecting more gratitude. The psychometric properties of the GQ-6 have been validated in a French population, with a Cronbach’s alpha of 0.74 [31]. To ensure methodological alignment with the original English validation study, we adopted a similar normalisation procedure for scale scores, facilitating direct comparison between the two versions.

### 2.4. Statistical Analysis

The study population was described with means and standard deviations for quantitative variables after normality testing using the Shapiro–Wilk test. Subsequent data analysis was conducted using R software (version 4.4.0) with a bilateral approach and a significance level of 0.05. The *psych* package (version 2.5.3) (Available online: https://ruby-doc.org/stdlib-2.5.3/libdoc/psych/rdoc/Psych.html (accessed on 1 March 2025)) was used for parallel and exploratory factor analysis, while the *lavaan* R package (version 0.6-19) (Available online: https://cran.r-project.org/web/packages/lavaan/index.html (accessed on 1 March 2025)). was used for confirmatory factor analysis. The various psychometric properties of the scale were tested as follows:

#### 2.4.1. External Validity (Hypothesis Testing)

Scale scores for the OAS-24, OAS-12, TOSCA-3-SS, RSES, MAAS, GQ-6 and SOP-MPS were normalised such that the lowest score was always zero and thereby corresponding to negative Likert endpoint descriptors such as “not at all”, “nothing” or “never” in the scales employed. This step was also taken to facilitate a comparison of means and standard deviations with corresponding scales used in the original English language validation of the OAS [18]. Relationships between scores were assessed using Spearman or Pearson correlation coefficients, depending on the normality of the distributions. Missing data were not substituted.

#### 2.4.2. Structural Validity

To ascertain the structural validity of the OAS-24 and OAS-12 items, a confirmatory factor analysis (CFA) was conducted. Model fit was evaluated using multiple indices, including χ^2^/df, the Tucker–Lewis index (TLI), comparative fit index (CFI), root mean square error of approximation (RMSEA), and standardised root mean squared residual (SRMR). The structure is adapted if χ^2^/df < 2; TLI and CFI were considered good if >0.90 and acceptable between 0.80 and 0.90; RMSEA and SRMR indicated a close model fit if ≤0.05 and a reasonable fit between 0.05 and 0.08. When CFA fails to adequately capture the underlying structure, an exploratory factor analysis (EFA) is conducted as a follow-up analysis.

For the EFA, Horn’s parallel analysis was employed, offering a more stringent criterion for factor retention compared to the Kaiser criterion and reducing the risk of over-extracting factors. This approach is particularly beneficial when exploring new factor structures in a different cultural context. For this parallel analysis, 50 simulations were generated to determine the optimal number of factors to retain [32]. The orthogonal varimax rotation was selected, and we employed factor analysis using minimising residuals (MINRES) to correct for any erroneous estimates [33].

#### 2.4.3. Reliability (Internal Consistency)

Item-total correlations, Cronbach’s alpha and Omega were calculated with the R packages *EFA.dimensions* (version 0.1.8.4) (Available online: https://www.r-pkg.org/pkg/EFA.dimensions (accessed on 1 March 2025)). and *psych* (version 2.5.3). The Omega figure, although less popular, is more reliable than Cronbach’s alpha to quantify the amount of random measurement error [34,35]. The Omega statistic was included as a complementary measure of internal consistency, providing a more nuanced estimation of reliability, especially in instances where the assumptions of tau-equivalence are not met.

#### 2.4.4. Reliability (Test–Retest)

Data for the test and retest conditions of the OAS were analysed using the intraclass correlation coefficient (ICC) and its 95% confidence interval (CI).

## 3. Results

### 3.1. Descriptive Characteristics (Validity, Sample Representativeness and Data Quality)

Of the 492 adult patients who enrolled in the study, 115 were males and 377 were females (sex ratio: 0.305). The mean age for males was 38.7 years (SD = 12.3; range = 18–71 years) and the mean age for females was 37.6 years (SD = 12.9; range = 18–79 years). All 492 participants were diagnosed with an emotional disorder, as the Therapy Center for Mood and Emotional Disorders at CHU Montpellier exclusively treats individuals with such conditions. However, due to incomplete data, the specific type of emotional disorder was not available for a portion of the sample. In addition to an emotional disorder, 44% of participants were also diagnosed with borderline personality disorder, while 3% had comorbid diagnoses of both borderline personality disorder and bipolar disorder. Conversely, 7% were identified as not having borderline personality disorder, and for the remaining 50%, the clinician recorded “I do not know if the person has a borderline personality disorder”.

Descriptive statistics for all measures taken are shown in Table 3. TOSCA (shame) scores showed significant skewness and, therefore, correlations of all measures employed Spearman’s tests.

### 3.2. Structural Validity

To evaluate the adequacy of the originally proposed six-factor structure based on Griffiths’ six components of addiction [20], a confirmatory factor analysis (CFA) was first conducted. As shown in Table 4, the model fit indices indicated a suboptimal fit. For the OAS-24 model, the chi-square test was significant (χ^2^(237) = 927), and the CFI (0.80) and TLI (0.77) did not reach the commonly accepted threshold of 0.90, suggesting that the predefined structure did not adequately represent the data. The RMSEA (0.08) was at the upper limit of acceptability, and the SRMR (0.07) suggested moderate misfit. Additionally, the covariances between latent factors were notably high, indicating a redundancy between certain dimensions and questioning the distinctiveness of the proposed factors.

A similar pattern was observed for the OAS-12 model (χ^2^(39) = 122), which, despite slightly better fit indices (CFI = 0.94, TLI = 0.90, RMSEA = 0.07), still presented issues related to item loadings (e.g., E4 and S4). Figure 1 summarised the results on standardised factor loadings, and inter-factor covariance for the OAS-24 and the OAS-12.

Given the limitations of the six-factor structure, we conducted an exploratory factor analysis (EFA) to empirically determine the most appropriate factor structure. The factor structure and variance are detailed in Table 5. Examination of the scree plots (Figure 2) suggests a four-factor solution based on the point at which the EFA scree plot lines meet those of the baseline. We explored two-factor, three-factor and four-factor solutions, and retained the three-factor solutions (see Appendix A).

Loadings of the factors are presented in Table 6 for the three-factor solution (and in Appendix B for the two- and four-factor solutions). Table 6 also shows item-total correlations, Cronbach’s alpha and Omega values. Through examining the factor loadings, it was clear that a single interpretable factor predominated (Factor 1), with generally high, positive loadings. The mean factor loadings for this were 0.37, suggesting that the sample size was adequate for the analysis [36]. This single factor, accounting for 18.5% of the variance, was deemed to represent the central construct of ontological addiction. Factor 2 appeared for the most part to reflect the material wealth/possessions dimension of the items. For example, high positive loadings were evidenced on S3 (“Thought about increasing or protecting your wealth or material possessions?”) and E3 (“Felt uplifted when you experienced financial or material gain?”). This factor accounted for 9.9% of the variance. The final third factor accounted for 8.1% of the variance although the provenance of this was less clear.

Conversely, low positive loadings were observed for S4 (“Thought about how you could avoid experiencing discomfort?”), E4 (“Felt good when you experienced fewer challenges?”), and W4 (“Found it hard to live more simply?”). The reasons for these low-factor loadings remain unclear. They could stem from issues related to item design, potential mistranslations, or conceptual ambiguity. Notably, two of these items (S4 and E4) belong to the “Sensations” category (see Table 1). This suggests that respondents may have encountered difficulties in consistently interpreting and responding to items related to their sensory experiences. Such inconsistencies would not be unexpected in a population characterised by emotional disorders, where the perception and regulation of sensory experiences are often altered.

The exploratory factor analysis indicated a predominantly single-factor solution, echoing findings from the original English language validation study. The single factor, capturing the essence of ontological addiction, consolidated items that relate to the core concept of self-representation and its maladaptive significance. This factor encapsulates the tendency to overidentify with certain self-aspects, leading to rigid self-perceptions and behaviours. It accounted for a significant portion of the data variance, suggesting that, similar to the English version, the French adaptation of the OAS is a robust tool for measuring this unidimensional construct.

### 3.3. Reliability (Internal Consistency)

The Cronbach’s alpha (α = 0.89) and Omega (ω = 0.81) figures of the OAS-24 showed good internal consistency and reflected the predominance of a single factor. The 12 items of the OAS-12 were selected in order to match the same 12 items as per the OAS-12 English language version of the scale [18]. In the present study, the internal consistency of the OAS-12 was good (α = 0.81; ω = 0.70), even though it was lower than that of the OAS-24 (α = 0.89; ω = 0.81), consistent with the substantially reduced number of items.

### 3.4. Reliability: Test–Retest

An important measure of validity is how well test and retest scores agree with one another across a particular time interval. Reliability (ICC) of the OAS-24 was 0.89 (CI: 0.81–0.94), which can be considered to be very good. Reliability (ICC) of the OAS-12 was also 0.89 (CI:0.81–0.94).

### 3.5. External Validity

Correlations between measures are detailed in Table 7, with all correlations reaching statistical significance at the *p* < 0.001 level. The results confirmed our hypotheses regarding the direction and strength of the correlations between OA and various established psychological constructs.

As expected, the Rosenberg Self-Esteem Scale (RSES), which measures global self-esteem, showed a strong negative correlation with OA (*r* = −0.484 for the OAS-24 and *r* = −0.52 for the OAS-12). This negative correlation underscores the idea that individuals with high levels of OA, driven by a perceived lack of worth in their self-concept, tend to struggle with low self-esteem.

Also, in line with our hypotheses, a significant positive correlation was observed between OA and the shame-proneness subscale of the Test of Self-Conscious Affect-3 (TOSCA-3). Shame, as a core emotion reflecting perceived inadequacy and the failure to meet the standards set by others, was strongly associated with OA, with correlations of *r* = 0.474 for the OAS-24 and *r* = 0.52 for the OAS-12. This underscores the link between OA and a heightened tendency to experience shame, further emphasising the detrimental impact of a flawed self-concept on emotional well-being.

Ontological addiction was also anticipated to correlate positively with the Self-Oriented Perfectionism subscale of the Multidimensional Perfectionism Scale (SOP-MPS), which reflects the internal drive for perfection and self-criticism. Consistent with this expectation, correlations were *r* = 0.400 for the OAS-24 and *r* = 0.402 for the OAS-12, supporting the notion that individuals with a high degree of OA are more likely to exhibit perfectionistic tendencies arising from a flawed self-concept.

Additionally, OA was predicted to negatively correlate with the Mindful Attention Awareness Scale (MAAS). The negative correlations observed (*r* = −0.446 for the OAS-24 and *r* = −0.435 for the OAS-12) underscore the incompatibility between the rigid self-concept inherent in ontological addiction and the open, non-judgmental awareness cultivated by mindfulness. This suggests that individuals with high OA scores may struggle to remain present and accepting, as they are preoccupied with satisfying a flawed self-concept.

Lastly, the Gratitude Questionnaire-6 (GQ-6), which assesses the propensity to recognise and appreciate positive aspects of life, was expected to negatively correlate with OA. This was confirmed by the observed correlations of *r* = −0.287 for the OAS-24 and *r* = −0.276 for the OAS-12, indicating that higher levels of OA are associated with a diminished capacity for gratitude. This finding illustrates that the need to maintain an incorrect self-concept may prevent individuals from appreciating the positive aspects of their lives.

### 3.6. Comparison of Psychometric Properties of OAS-24 and OAS-12

The OAS-24 demonstrated good internal consistency with an α of 0.89 and ω of 0.81. In contrast, the OAS-12 showed a reduction in these values to α = 0.81 and ω = 0.70, respectively. However, this reduction will inevitably reflect the fewer number of items and it should be noted that the OAS-12 still maintains satisfactory internal consistency. This makes the shorter OAS-12 a suitable and pragmatic alternative to the OAS-24 given its efficiency and reduced participant burden.

## 4. Discussion

The present study aimed to validate the French version of the Ontological Addiction Scale (OAS), an instrument designed to measure the severity of dysfunctional ego-centredness. The French OAS demonstrated excellent internal and convergent validity, with the latter supported by negative correlations with established measures for self-esteem, mindfulness and gratitude. Furthermore, positive correlations were found between the OAS and established measures of shame and self-orientated perfectionism. These findings are consistent with theoretical expectations and reinforce the scale’s construct validity, indicating that higher levels of OA are associated with markers for poor mental health, low life satisfaction and maladaptive interpersonal relationships. These findings mirror those in the original English validation study [18] as well as a subsequent cross-sectional correlation study using the English version of the scale [37]. This alignment supports the cross-cultural applicability of the OAS and its relevance in measuring ontological addiction across different languages and populations.

The EFA and parallel analysis suggested a three-factor solution. However, upon examining the items loading onto each factor (see Appendix A), we found no clear semantic coherence uniting all items within each factor. This raised concerns about content validity, leading us to conclude that the three-factor solution was not sufficiently robust for recommendation. Consequently, we advocate for the single-factor structure, which is more interpretable and aligns better with previous findings.

Furthermore, test–retest reliability confirmed the stability of the French OAS over a 15-day interval, with significant correlations for both the OAS-24 and OAS-12 versions. This stability is crucial for the scale’s application in both clinical and research settings, ensuring that it reliably measures the construct over time.

## 5. Clinical Implications

Ontological addiction, characterised by the attachment to a flawed self-concept, is hypothesised to be at the root of many psychiatric disorders. This aligns with the broader movement in psychiatry to transcend the limitations of categorical diagnostic systems, in order to better capture the continuous and dimensional nature of psychopathological constructs [38]. Similarly, the Research Domain Criteria initiative by the National Institute of Mental Health [39] represents a significant shift towards understanding mental disorders through dimensions of observable behaviour and neurobiological measures. This approach underscores the importance of constructs such as ontological addiction, which cut across traditional diagnostic boundaries and provide a more nuanced understanding of mental health.

The cybernetic perspective on psychopathology further supports this shift [38], which conceptualises mental illness not merely as statistical deviance or brain disease but as a failure in goal-directed behaviour and adaptation. From this viewpoint, psychopathology arises when individuals are unable to modify their goals or strategies in response to changing circumstances, leading to persistent dysfunction. The concept of ontological addiction fits well within this cybernetic framework and highlights how maladaptive self-identifications can lead to a wide range of psychiatric symptoms and behaviours. By focusing on the root issue—addiction to an incorrect self-concept—clinicians are better equipped to address the underlying causes of various mental health issues rather than just their symptoms. This paradigm shift is crucial as traditional diagnostic categories often fail to capture the complexity and fluidity of mental health conditions. Therefore, as a central clinical dimension, OA can offer a more integrative and dynamic approach to understanding and treating mental disorders.

Thus, the OAS could be used to identify individuals who are particularly susceptible to ontological addiction, guiding personalised therapeutic interventions. Techniques from third-wave cognitive-behavioural therapies and Buddhist-derived practices, such as mindfulness meditation, loving-kindness meditation [40], compassion meditation [41] or deconstructive meditation [42], may be particularly effective in this respect. These approaches aim to deconstruct familiarity with incorrect self-concepts and promote a healthier, more adaptive sense of self. Indeed, research has shown that meditative interventions targeting ontological addiction can lead to significant improvements in psychological well-being [43]. For instance, meditation-based therapies have been effective in reducing symptoms of depression, anxiety, and other psychiatric conditions by fostering a more flexible and accurate self-identification [42].

## 6. Limitations and Future Directions

Despite the promising findings, the present study is limited by a number of factors. The generalisability of the results may be limited given that recruitment focused on individuals with specific mental health issues, although this was mitigated to a certain extent by participants representing a wide range of sociodemographic backgrounds. The relatively short test–retest interval of 15 days was an additional potentially limiting factor, which could be extended in future studies to provide a better understanding of the scale’s stability over longer periods. It is also recommended that future studies involve larger and more varied clinical populations to confirm the scale’s applicability and robustness.

Future studies should also assess the suitability of the OAS as a direct measure of progress in psychotherapy, particularly for mindfulness and other third-wave cognitive-behavioural therapies (CBTs), which often target ontological addiction components. Finally, understanding the neural and psychological mechanisms underlying ontological addiction will also be crucial. Such enquiry could involve integrating findings from neuroscience with psychometric data to develop more comprehensive models of mental health and illness.

## 7. Conclusions

The present study showed that the French version of the OAS demonstrated strong reliability and validity, aligning with the psychometric properties of the original English language version [18]. Both the French and English language versions of the OAS appear to reflect a robust measure of ontological addiction, suitable for clinical assessment and research. Future studies should continue to refine the scale and explore its applicability in additional cultural contexts, ultimately enhancing our understanding of ontological addiction and its impact on mental health. The present French language validation of the OAS also contributes to assessment efforts orientated towards a more dimensional understanding of psychopathology and associated interventions. As Ducasse et al. [7] noted, “One should realise that if the understanding of Self-identification is impaired, all the interventions implemented to decrease the Self’s suffering will be subsequently impaired.” This underscores the importance of wisdom-informed perspectives around self-concept in psychiatric care to prevent perpetuating suffering. The OAS thus holds potential not only for tracking therapeutic progress but also for improving the understanding and treatment of self-related dysfunctions in mental health.

## Figures and Tables

**Figure 1 ijerph-22-00511-f001:**
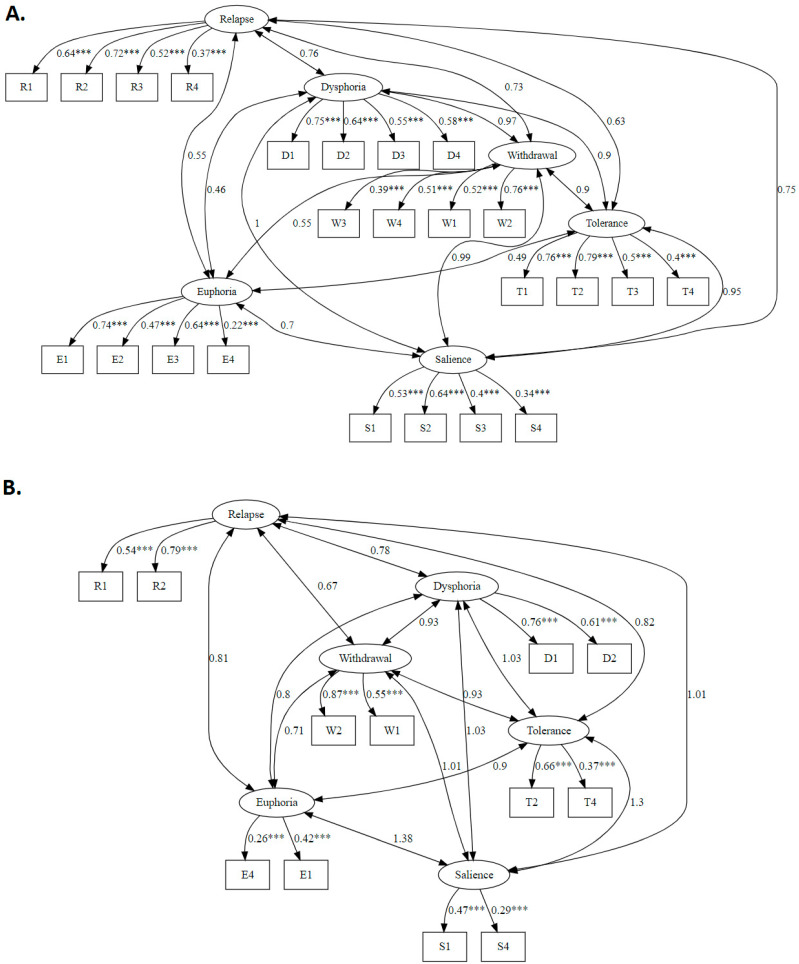
Confirmatory factor analysis for the OAS-24 (**A**) and OAS-12 (**B**). Standardised factor loadings and covariances are displayed, with asterisks indicating the significance of latent relationships (*** *p* < 0.001).

**Figure 2 ijerph-22-00511-f002:**
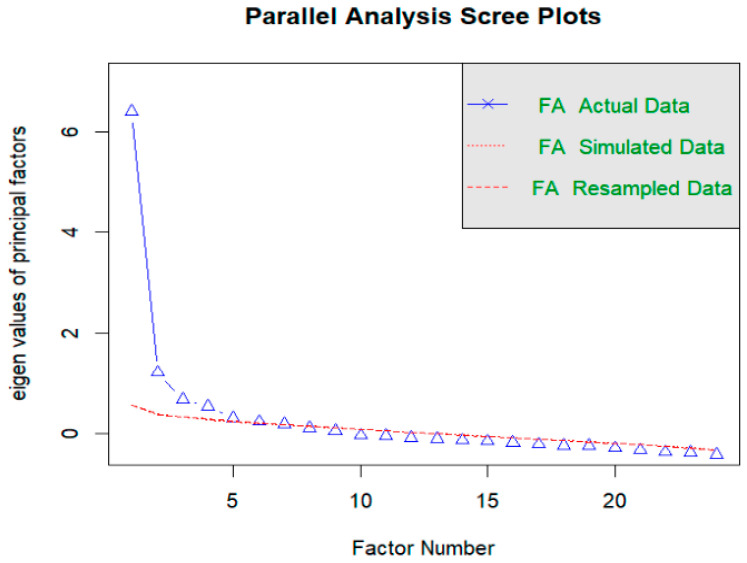
Parallel forms analysis of factors extracted from OAS-24 responses. The number of triangles above the lower dotted line is the number of suitable factors. Factor analysis (FA) Simulated data (dotted line) and FA resampled (dashed line) data are stacked.

**Table 1 ijerph-22-00511-t001:** Four-factor breakdown of the *Eight Mundane Concerns* of Buddhism.

Component	Positive	Negative
1. Material wealth/possessions	Delight about gain	Distress about loss or non- acquisition
2. Sensations	Delight at pleasurable sensations	Distress from painful or unpleasant sensations
3. Others’ perception of us (friends, colleagues, family)	Delighted with praise	Distress from disapproval or criticism
4. Others’ perception of us (general reputation)	Delighted with good reputation	Distress from bad reputation

**Table 2 ijerph-22-00511-t002:** Ontological Addiction Scale (OAS-24): six subscales based on Griffiths’ [20] component model of addiction. Presented in bold, the 12 items of the OAS-12.

**Salience (i.e., ego-centred activities are the most important in the person’s life, and dominate their thinking, feelings and behaviours)**	**S1. Felt you needed to receive more attention or affection from a person you care about?** **S2. Thought about how others see you?** **S3. Thought about increasing or protecting your wealth or material possessions?** **S4. Thought about how you could avoid experiencing discomfort?**
**Euphoria (i.e., ego-centred occurrences impact mood in a positive way)**	**E1. Felt uplifted when you were praised?** E2. Felt superior to others? E3. Felt uplifted when you experienced financial or material gain? **E4. Felt good when you experienced fewer challenges?**
**Tolerance (i.e., one needs to constantly increase ego-centred behaviour to feel well)**	T1. Felt you needed to try harder in order to receive praise or avoid criticism? **T2. Felt you needed to do better in order to avoid shame or humiliation?** T3. Felt you needed more money or material possessions? **T4. Felt an increasing need to occupy yourself to avoid being on your own?**
**Withdrawal (i.e., unpleasant feeling occurs when ego-centred behaviour is reduced)**	**W1. Found it hard to accept your mistakes and shortcomings?** **W2. Found it hard to overcome rejection?** W3. Found it hard to give something away? W4. Found it hard to live more simply?
**Dysphoria (i.e., interpersonal or intrapsychic conflicts resulting from ego-centred behaviour)**	**D1. Felt low when you were criticised?** **D2. Felt inferior to others?** D3. Felt low when you encountered financial or material loss? D4. Felt low when you encountered difficult circumstances?
**Relapse (i.e., the tendency for repeated reversions to ego-centeredness following a period of being less self-centred)**	**R1. Stopped being kind to somebody you care about because they criticised you? *** **R2. Felt worried about not being recognised after having acted in others’ interests?** R3. Felt regret after having given a gift? R4. Stopped helping others because it was causing discomfort? **

Note: Two items were adapted from the English version to better fit the French cultural context: * “Stopped being kind to somebody you care about because they offended you?” ** “Stopped helping others because it was causing discomfort or inconvenience”.

**Table 3 ijerph-22-00511-t003:** Descriptive statistics (*n* = 492; 115 males, 377 females).

Measure	*n*	Mean	SD	95% Confidence Interval	
				Lower Bound	Upper Bound
**Age**	489	37.7	12.9	36.6	38.9
**OAS-24**	453	53.6	14.0	52.3	5.9
**OAS-12**	466	30.3	7.5	29.6	30.9
**TOSCA-3-SS**	433	36.9	11.7	35.8	38.0
**RSES**	451	13.3	6.4	12.7	13.9
**MAAS**	435	38.1	13.0	36.9	39.3
**GQ-6**	489	20.5	8.01	19.8	21.2
**SOP-MPS**	437	56.4	17.3	54.7	58.0

**Table 4 ijerph-22-00511-t004:** Results for the confirmatory factor analysis.

Model	χ^2^	df	χ^2^/df	TLI	CFI	RMSEA	SRMR	Coeff > 0.05	Loading < 0.30
OAS-24	927	237	3.9	0.77	0.80	0.08	0.07	–	E4
OAS-12	122	39	3.1	0.90	0.94	0.07	0.04	–	S4, E4

χ^2^ = chi^2^ test; df = degrees of freedom; χ^2^/df = chi^2^ test divided by degrees of freedom; TLI = Tucker–Lewis Index of Fit; CFI = comparative fit index; RMSEA = root mean square error of approximation; SRMR = standardised root mean squared residual. Loading: standardised loadings. *p* < 0.001 for all models.

**Table 5 ijerph-22-00511-t005:** Exploratory factor analysis: factor structure of OAS-24. Prototype-based on initial extraction of items with eigenvalue > 1.

Factor	Total	% of Variance	Cumulative %
1	7.17	30	30
2	1.94	8	38
3	1.46	6	44
4	1.33	6	50
5	1.07	4	54
6	1.03	4	58

**Table 6 ijerph-22-00511-t006:** Factor loadings and corrected item-total correlations (CITCs) of OAS-24 group scores for the three-factor solutions. Factor loadings > 0.3 are in bold.

Item Groups: *Salience* (S1-7), *Ego Euphoria* (E1-5), *Tolerance* (T1-5), *Dysphoria* (D1-4), *Withdrawal* (W1-4) and *Relapse* (R1-6)	OAS-24 (α = 0.89; ω = 0.81)			
	Factor 1	Factor 2	Factor 3	CITC
**S1. Felt you needed to receive more attention or affection from a person you care about?**	**0.455**	0.119	0.270	0.517
**S2. Thought about how others see you?**	**0.609**	0.200	0.129	0.602
**S3. Thought about increasing or protecting your wealth or material possessions?**	0.074	**0.631**	0.238	0.479
**S4. Thought about how you could avoid experiencing discomfort?**	0.254	0.200	0.137	0.361
**E1. Felt uplifted when you were praised?**	0.218	0.241	**0.385**	0.473
**E2. Felt superior to others?**	−0.012	0.130	**0.456**	0.273
**E3. Felt uplifted when you experienced financial or material gain?**	0.159	**0.476**	0.279	0.488
**E4. Felt good when you experienced fewer challenges?**	0.153	0.218	0.146	0.294
**T1. Felt you needed to try harder in order to receive praise or avoid criticism?**	**0.676**	0.180	0.127	0.641
**T2. Felt you needed to do better in order to avoid shame or humiliation?**	**0.717**	0.273	0.044	0.675
**T3. Felt you needed more money or material possessions?**	0.224	**0.786**	0.048	0.562
**T4. Felt an increasing need to occupy yourself to avoid being on your own?**	**0.303**	0.162	0.215	0.407
**W1. Found it hard to accept your mistakes and shortcomings?**	**0.461**	0.119	0.110	0.448
**W2. Found it hard to overcome rejection?**	**0.707**	0.159	0.206	0.684
**W3. Found it hard to give something away?**	0.182	0.292	**0.323**	0.439
**W4. Found it hard to live more simply?**	0.352	0.363	0.188	0.538
**D1. Felt low when you were criticised?**	**0.687**	0.131	0.261	0.683
**D2. Felt inferior to others?**	**0.675**	0.192	−0.053	0.559
**D3. Felt low when you encountered financial or material loss?**	**0.317**	**0.620**	0.197	0.625
**D4. Felt low when you encountered difficult circumstances?**	**0.568**	0.116	0.121	0.524
**R1. Stopped being kind to somebody you care about because they offended you?**	0.282	0.144	**0.544**	0.529
**R2. Felt worried about not being recognised after having acted in others’ interests?**	**0.502**	0.055	0.442	0.591
**R3. Felt regret after having given a gift?**	0.128	0.216	**0.505**	0.441
**R4. Stopped helping others because it was causing discomfort or inconvenience?**	0.070	0.062	**0.436**	0.294

**Table 7 ijerph-22-00511-t007:** Correlation matrix of measures taken (Spearman’s Rho).

Measure	OAS-24	OAS-12	TOSCA-3 -SS	RSES	MAAS	GQ-6	SOP-MPS
**OAS-24**	1	0.927	0.475	−0.513	−0.449	−0.294	0.4
**OAS-12**	0.927	1	0.501	−0.53	−0.434	−0.271	0.402
**TOSCA-3-SS**	0.475	0.501	1	−0.551	−0.406	−0.218	0.37
**RSES**	−0.513	−0.53	−0.551	1	0.397	0.444	−0.284
**MAAS**	−0.449	−0.434	−0.406	0.397	1	0.205	−0.358
**GQ-6**	−0.294	−0.271	−0.218	0.444	0.205	1	−0.092
**SOP-MPS**	0.4	0.402	0.37	−0.284	−0.358	−0.092	1

All correlations significant to *p* < 0.001; OAS-24—Ontological Addiction Scale; OAS-12—Ontological Addiction Scale (short-form); TOSCA-3-SS—Shame Subscale of Test of Self-Conscious Affect-3 Scale; RSES—Rosenburg Self-Esteem Scale; MAAS—Mindful Attention Awareness Scale; GQ-6—Gratitude Scale Questionnaire; SOP-MPS—Self-Oriented Perfectionism Subscale of the Multidimensional Perfectionism Scale.

## Data Availability

Anonymised data from the study will be made publicly available upon notification of acceptance for publication.

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
