# Peer review of "French Translation and Validation of the Ontological Addiction Scale (OAS)"

_ijerph, 2025, doi:10.3390/ijerph22040511_

Round 1
Reviewer 1 Report
Comments and Suggestions for Authors
Thanks to the authors for sharing their manuscript. I think the scale adaptation is well described, but there are important points to consider.
The main comment concerns the evaluation of the factor structure. The authors describe in detail the results of the exploratory factor analysis, but it remains unclear why they did not perform a confirmatory factor analysis. It seems to me that the inclusion of the results of confirmatory factor analysis in the psychometric analysis will significantly improve the evidence of the validity of the adapted instrument.
Further, it seems doubtful to call the results of evaluating the factor structure of the scale using exploratory factor analysis evidence of structural validity. Rather, we are talking about factor validity as a kind of structural validity.
Another inaccuracy in the use of terms is seen in the description of internal reliability. So, the authors write about “excellent internal consistency (Cronbach's alpha: 0.89)” and “satisfactory internal consistency (Cronbach's alpha: 0.81)”. Meanwhile, Cronbach's alpha coefficients from 0.8 to 0.9 are considered good (excellent at more than 0.9, satisfactory at 0.7-0.8).
In general, I believe that by adding the results of the confirmatory factor analysis and clarifying the terms, the manuscript can be recommended for publication.
Sincerely yours,
the reviewer.
Author Response
Thanks to the authors for sharing their manuscript. I think the scale adaptation is well described, but there are important points to consider.
The main comment concerns the evaluation of the factor structure. The authors describe in detail the results of the exploratory factor analysis, but it remains unclear why they did not perform a confirmatory factor analysis. It seems to me that the inclusion of the results of confirmatory factor analysis in the psychometric analysis will significantly improve the evidence of the validity of the adapted instrument.
Further, it seems doubtful to call the results of evaluating the factor structure of the scale using exploratory factor analysis evidence of structural validity. Rather, we are talking about factor validity as a kind of structural validity.
Response: Thank you for raising these important points. A confirmatory factor analysis has now been performed. The text for sections 2.4.2 and 3.2 has been revised accordingly, and a new table (Table 4) and figure (Figure 1) added to fully describe and illustrate the findings of this analysis.
Another inaccuracy in the use of terms is seen in the description of internal reliability. So, the authors write about “excellent internal consistency (Cronbach's alpha: 0.89)” and “satisfactory internal consistency (Cronbach's alpha: 0.81)”. Meanwhile, Cronbach's alpha coefficients from 0.8 to 0.9 are considered good (excellent at more than 0.9, satisfactory at 0.7-0.8).
Response: Thank you for pointing out these lapses in the precise use of language to describe ranges of Cronbach’s alpha values. This has now been rectified (see lines 26, 28, 366, 369, and 418).
In general, I believe that by adding the results of the confirmatory factor analysis and clarifying the terms, the manuscript can be recommended for publication.
Response: We agree, and have completed these changes as described above.
We hope that the revisions detailed above are satisfactory.
Reviewer 2 Report
Comments and Suggestions for Authors
I find it particularly interesting and important to study samples of people with diverse problems, in this case with mood disorders, since they are the ones who require effective psychological interventions.
However, I find it problematic that the authors have not tested the original structure and some other alternative using Confirmatory Factor Analysis.
Conclusions such as the following can only be made if Confirmatory Factor Analysis is performed: “The EFA and parallel analysis pointed towards a three-factor solution, although the single-factor structure was most interpretable and consistent with previous findings.”
I also find the description of the problem presented by the sample to be imprecise: what does it mean that “7% did not have a diagnosis of borderline personality disorder,” and why can the authors affirm that the entire sample has a disorder and not know what type?
Some explanation should be given for the low factor weight of some items, such as “E4. Felt good when you experienced fewer challenges?” or “W4. Found it hard to live more simply?”
Between Table 2 and 3 there is one without numbering or any information (page 8)
Author Response
I find it particularly interesting and important to study samples of people with diverse problems, in this case with mood disorders, since they are the ones who require effective psychological interventions.
However, I find it problematic that the authors have not tested the original structure and some other alternative using Confirmatory Factor Analysis.
Conclusions such as the following can only be made if Confirmatory Factor Analysis is performed: “The EFA and parallel analysis pointed towards a three-factor solution, although the single-factor structure was most interpretable and consistent with previous findings.”
Response: Thank you for raising these important points. A confirmatory factor analysis has now been performed. The text for sections 2.4.2 and 3.2 has been revised accordingly, and a new table (Table 4) and figure (Figure 1) added to fully describe and illustrate the findings of this analysis.
I also find the description of the problem presented by the sample to be imprecise: what does it mean that “7% did not have a diagnosis of borderline personality disorder,” and why can the authors affirm that the entire sample has a disorder and not know what type?
Response: Thank you for raising this point. We have elaborated upon the characteristics of the sample recruited, providing as comprehensive as possible a description and detailing the limitations of the data provided to the researchers (see lines 268-277).
Some explanation should be given for the low factor weight of some items, such as “E4. Felt good when you experienced fewer challenges?” or “W4. Found it hard to live more simply?”
Response: Thank you for this suggestion. We have added a section in which we briefly examine these findings and explore possible reasons for them (see lines 316-325).
Between Table 2 and 3 there is one without numbering or any information (page 8)
Response: All tables and figures are now properly headed.
We hope that the revisions detailed above are satisfactory.